# Design for 3D Printed Tools: Mechanical Material Properties for Direct Polymer Additive Tooling

**DOI:** 10.3390/polym14091694

**Published:** 2022-04-21

**Authors:** Peter Frohn-Sörensen, Michael Geueke, Bernd Engel, Bernd Löffler, Philipp Bickendorf, Arian Asimi, Georg Bergweiler, Günther Schuh

**Affiliations:** 1Forming Technology, Institute of Production Technologies, University of Siegen, 57076 Siegen, Germany; michael.geueke@uni-siegen.de (M.G.); bernd.engel@uni-siegen.de (B.E.); 2Laboratory for Machine Tools and Production Engineering (WZL) of RWTH Aachen University, 52074 Aachen, Germany; p.bickendorf@wzl.rwth-aachen.de (P.B.); arian.asimi@rwth-aachen.de (A.A.); g.bergweiler@wzl.rwth-aachen.de (G.B.); g.schuh@wzl.rwth-aachen.de (G.S.)

**Keywords:** additive tooling, rapid tooling, additive manufacturing, FFF, FDM, polymers, flexibility, metal forming, mass customization

## Abstract

In relation to the fourth industrial revolution, traditional manufacturing methods cannot serve the flexibility demands related to mass customization and small series production. Rapid tooling provided by generative manufacturing has been suggested recently in the context of metal forming. Due to the high loads applied during processes to such tooling, a purposeful mechanical description of the additively manufactured (AM) materials is crucial. Until now, a comprehensive characterization approach for AM polymers is required to allow a sophisticated layout of rapid tooling. In detail, information on compressive and flexural mechanical properties of solid infilled materials made by additive manufacturing are sparsely available. These elementary mechanical properties are evaluated in the present study. They result from material specimens additively manufactured in the fused filament fabrication (FFF) process. The design of the experiments reveals significant influences of the polymer and the layer height on the resulting flexural and compressive strength and modulus as well as density, hardness, and surface roughness. As a case study, these findings are applied to a cup drawing operation based on the strongest and weakest material and parameter combination. The obtained data and results are intended to guide future applications of direct polymer additive tooling. The presented case study illustrates such an application and shows the range of manufacturing quality achievable within the materials and user settings for 3D printing.

## 1. Introduction

Due to the transformation from mass production to mass personalized production in the last decades, this paradigm shift requires alternative production techniques and manufacturing processes to fulfil the individualized consumer demand [1]. Mass personalized product individualization, shortened product lifecycles and lead times, as well as increased product derivatives, necessitate economic alternatives for traditional production techniques, and manufacturing processes typically implemented for great lot sizes [2,3,4]. Steadily tightening political and environmental regulations paired with globalization and price competition forces manufacturers to reconsider their strategies on the distribution markets [5]. When it comes to cost and time consumption in large-scale industrial production, alternative approaches are inevitable to ensure transformation from mass production to mass personalized production.

Sheet metal forming, as one of the common metal forming techniques, is an omnipresent production process for automotive, naval, and aerospace, as well as household appliance, typically aiming for great lot size production due to cost intensive tool investments and a high level of automation [6].

On the other hand, mass personalized production requires agile reconfigurability to ensure a fast, individualized, and economic process for small lot size, shape complexity, and physical parameter variation [7].

Since there are no suitable and economically approved techniques to ensure individualized and flexible small batch size production in deep drawing, further approaches are required to fulfil these agile demands [2]. Additive manufacturing (AM) may empower an affordable alternative to tackle high complexity in reconfigurable production systems, since it enables a flexible production for individualized products, neglecting cost penalties in manufacturing [8]. In AM, 3D data is transformed directly into physical parts without further tooling. Disregarding the comparable high production time of AM processes as an ongoing technological hard and software improvement, most of the AM processes are material-saving. The AM process generates objects layer-wise on demand, whereas traditional manufacturing like grinding or milling are subtractive processes. Therefore, direct polymer additive tooling (DPAT) is a promising tooling method that may be used in sheet metal forming for small batch size production and prototyping [9].

This work aims to study the mechanical performance of four FFF materials, namely polylactic acid (PLA), polycarbonate (PC), polyamide (PA), and polyethylene terephthalate glycol modified (PETG). Flexural and compression tests are performed in a full-factorial design of experiments. Subsequently, to investigate the performance of DPAT for deep drawing, the weakest and the strongest parameter results are applied to a cup drawing experiment using tooling made by the FFF process.

### Background and Related Work

Equipment for conventional production technologies can be manufactured with AM to enable small to medium-sized series in the original material. The production of components and tools with short lead time by means of additive manufacturing is characterized by a variety of terms. The most used terms are rapid tooling, rapid prototyping, and rapid manufacturing. With respect to AM and the production of manufacturing tools, these terms can also be summarized as additive tooling (AT). AT enables direct manufacturing of tools with close to final product quality [10]. Applications for AT as a production technology range from casting and injection molds to cutting and forming tools. Shorter lead times are a decisive advantage of AT and enable economical production of tools for small batches [11]. Particularly in the highly competitive automotive industry, it is possible to benefit from this advantage and achieve a shorter time-to-market [12]. In this context, forming tools manufactured by AT to produce car body parts have the potential to meet the requirements in terms of flexibility and ever shorter lead times. The production of three-dimensional sheet metal workpieces, such as those used in the car body, is usually carried out by deep drawing. Corresponding DPAT tools can achieve similar results to traditional sheet metal forming tools with limitations in the aspect of fatigue life [13].

In the deep drawing process, a sheet blank is restrained peripherally by one or multiple blank holder units. Afterwards, a punch draws the material radially into a forming die. This enables complex open hollow bodies, like cup or box contours, in single- or multiple-step iterations. Compared to stretch forming, where a considerable change of the sheet thickness is achieved, deep drawing allows the material to flow with the aid of a blank holder and, therefore, ideally aims for constant material thickness [14]. Since deep drawing applications usually aim for mass size production, grey casting accompanied by post-milling is the established way of tool manufacturing [15]. The application area of deep drawing tools is an important field of research due to the complex load conditions that push the tools and the sheet metal material to their limits [16]. Even though DPAT is widely used in embossing and bending operations as well as injection mold tooling, this application has only been used occasionally in sheet metal forming so far [17]. On the other hand, metal based additive tooling approaches have been studied more extensively. However, compared to conventional subtractive tooling methods, these techniques require high pre-investments and precise machinery, usually exceeding tool costs for small batch size and try-outs.

Some DPAT approaches for sheet metal forming have been studied by several authors. While [9,10,18] stated general insights, limitations, and design recommendations for DPAT, other researchers focused explicitly on practical try outs and feasibility approaches. Durgun [19] investigated a V-shaped tailgate forming tool made from polycarbonate for 101 forming operations on DC04 (0.8 mm) and S355MC (0.8 mm) sheet metal. For DC04, the tools showed good dimensional accuracy over the whole batch, whereas S355MC showed dimensional stability up to the 50th part. Prior to a feasibility study existing for two different drawing tools, Schuh et al. investigated the forming behavior of AM parts via cupping tests and structural infill optimization. In addition, they included a simulation approach and optical measurements of the forming tools to determine the deformation behavior up to 23 forming operations [16]. Nakamura et al. [20] examined the forming behavior of a V-bent and a deep drawn cylindrical cup for three different sheet metals. Besides the geometrical accuracy, they investigated the surface roughness after the forming operation. In addition, stiffening metal elements were inserted into the AM tool structures and therefore enhance their performance. Schuh et al. assessed the geometric accuracy of DPAT for PLA material on different geometry features in a deep drawing process. They tested a demonstrator for 27 drawing iterations, where they stated a stable tolerance of ±0.5 mm for a DC04 sheet metal part (1 mm) in four out of five geometry features [21].Aksenov and Kononov [22] studied the performance of DPAT on thin aluminum sheets via multiple V-bent operations of 2–4 mm height without a noticeable wear behavior on the polyethylene terephthalate (PET) tool after a few dozen forming operations. Frohn-Sörensen et al. [23] examined the performance and geometrical stability for DPAT on a rubber pad forming process for DC04 (0.7 mm) up to 64 parts. The results showed that the PLA dies could be used to draw conventional sheet metal with a stabilized performance after the 32nd forming operation. Geueke et al. [24] extended this approach and performed a topology optimization on the forming dies, where they reduced the AM material up to 30% with negligible geometrical forming deviations compared to the rigid die from the previous study. Bergweiler et al. [17] investigated the dimensional precision of deep drawn cups using DPAT with lot sizes up to 20 DC04 (1 mm) parts, where they stated a geometrical deviation of ±0.5 mm on the forming tools. Löffler et al. [25] examined deep drawing using DPAT and PLA material on DC04, aiming for reduction of tooling costs. They stated a general feasibility with geometrical deviations up to −0.93 mm on the tool set with cost reduction of 93% compared to conventional tooling.

Prior to DPAT, several authors conducted mechanical performance tests for different polymers and load types via AM. Since the geometries for tensile und flexural tests are clearly specified in the standards DIN EN ISO 527 and ISO 178, respectively, the compression test geometry according to EN ISO 604 refers to an inequation for different specimen shapes of pipes, cylinders, or prisms. This inconsistency leads to the issue of authors using non-uniform specimen contours. For example, Wang et al. [26] used a 36 mm cube of pure filler structure, whereas ref. [20] used a cylinder with 25 mm diameter and 30 mm height, which makes the test results hard to compare. Besides that, the compression tests by refs. [20,27] revealed different values for PLA material. Nakamura et al. [20] achieved 30 MPa, whereas ref. [27] gained a compression strength of 59.78 MPa for PLA specimen. A similar discrepancy can also be stated by the investigations of refs. [28,29], as they gained higher mechanical strength for polymer-based AM tensile specimen with higher layer thickness (0.3 mm), while ref. [30] obtained a better performance for smaller layer thickness (0.05 mm). This inconsistency can be further confirmed for compression test results with respect to layer thickness. Sood et al. [31] found out that higher layer thickness (0.254 mm) improved the mechanical performance, even though ref. [32] gained contrary results, as they improved the performance with smaller layer heights (0.14 mm). For flexural testing, this discrepancy revealed contrary test results. Sood et al. [33] could improve the performance with higher layer height (0.254 mm), although ref. [34] revealed better results for lower layer height (0.1 mm).

In general, a comparison between the aforementioned outcomes should be viewed with caution since the AM filaments and printing parameters were not used consistently and uniformly. Due to the partly inconsistent and contradictory mechanical test results for identical polymers, further investigations are necessary to predict and apply DPAT in sheet metal forming and the production sector in general. Even though literature reveals a large amount of tensile test data, fewer flexural and compression data are available. To ensure a performant application of DPAT in sheet metal forming, this study can contribute to this field of research [35].

## 2. Materials and Methods

### 2.1. Parameters and Properties for Additive Tooling

In the following, the essential properties and slicing parameters for additive tooling (AT) for forming technologies are derived, justified, and described. Subsequently, they are used to determine a suitable test setup for material characterization. The results are used to assess the process suitability as a forming tool material. Different forming technologies are distinguished according to their main stress type in the technical standards DIN 8582 (cf. Figure 1), Schuler GmbH [36] (p. 7), and DIN 8585-1 (p. 3; cf. Appendix A). This results from the introduced stresses, which are the outcome of the corresponding relative movements between the workpiece and the tool, cf. ref. [36] (pp. 6–18). In production technology, deep drawing, stamping, and bending are of particular importance.

Deep drawing is characterized by a combination of tensile and compressive stresses. In addition, shear stress results from the material flow of the sheet metal flowing downstream. The general stress spectrum is depicted in DIN 8585-4 (p. 3, see Appendix A). In contrast to deep drawing, stamping is dominated by tensile forming.

In the following, an idealized cup geometry is considered for a deep drawing process (cf. Figure 2), see [37] (p. 16) and [14] (p. 262). While uniaxial tensile stress dominates in the punch edge rounding at the sheet metal component, the bottom of the cup experiences a tensile–tensile stress. In the flange area, tensile stress is combined with compressive stress due to the often rotationally symmetrical shape of formed components.

Polymer-based additively manufactured tools exhibit an elastic–plastic deformation behavior and are more suitable for use areas with limited load/pressure due to their reduced mechanical properties and higher wear. They are more sensitive to stresses during deep drawing (cf. Figure 2) than conventional tools due to their lower strength and greater susceptibility to wear. Deep drawing tools are typically made of metal (especially free-cutting steel and aluminum), which are assumed to be quasi-static and exhibit low wear due to the use of lubricants. Thus, it is particularly important to adapt the design of the additive manufacturing process to the load cases of forming technologies in order to ensure a sufficient service life, cf. [21] (p. 4).

The stability of additively manufactured components depends in particular on the infill density. As a result of a higher infill density, an increase in compressive strength is observed. The infill pattern should determine the volume filling of the component as efficiently as possible, which can optimize material efficiency, production time, and costs, cf. [38] (pp. 16–18). The infill is based on a fully computer-generated and repetitive pattern. The diameter of the selected nozzle defines the range in which the layer height is adjusted in the slicing software. The thickness, as well as the number of wall lines, which is prioritized in most slicers, also influences the internal stability and strength of additively manufactured components. Therefore, the number of walls (perimeters) must be determined to gain a clean mapping of the contours. The temperature of the nozzle and the printing platform affect the strength of the connection between the material strand and the underlying layers. Usually, a material comes along with its individual data sheet, which is based on empirical try-out parameters for the appropriate system technology to produce a satisfactory print result. A high degree of overlap between the layers increases the internal stability of additively manufactured components, although the resulting over-extrusion may cause geometric deviations. The higher the printing speed, the faster and more favorably the manufacturing process can be designed. However, if the printing speed is too high, the layer adhesion is not sufficiently pronounced, so that the internal strength is reduced. Further on, higher printing speed comes along with higher acceleration and deceleration of the printing heads, which may influence the mechanical performance and lifespan of the printer. Additionally, the printing direction affects the performance of additively manufactured forming tools. Here, it is important that the layer-wise building direction lines up with the direction of the main load during the forming process. Otherwise, the probability of interlayer delamination increases.

Material parameters, which are decisive for the suitability as a tool material, are related in particular to conventional mechanical parameters. These are given in the technical datasheets (TDS) of the filament manufacturers. Unfortunately, they do not include the influence of the slicing parameters on the AM processes and machines. Often these values are determined according to DIN EN ISO 527(pp. 1–34, see Appendix A) with various assumptions (e.g., regarding the printing speed, filament overlay, infill density). The most important mechanical parameters considered in the application case of forming technology are the following:Modulus of elasticity or tensile modulus (given in most TDS) of additively manufactured specimen (in GPa);Tensile strength printed specimen (specified in most TDS) in one or more printing directions (in MPa);Compressive stress or compressive yield stress at 5% compression (not specified in TDS) (in MPa);Density of specimen (specified in most TDS) (in g/mm^3^);D-Shore-hardness of specimen (not specified in TDS).

The economic efficiency and service life of a tooling system is significantly influenced by the choice of the tool material and its pairing with the corresponding sheet metal. To confirm the suitability of these materials for the use as inserts in forming tools, it is necessary to investigate the strength properties in particular. In addition to various slicing parameters, batch variations in the filament production also have an influence. To reduce the probability of these effects occurring, all materials are requested from the same manufacturer at the same time. Following the delivery (vacuum packed), all material is used up for printing within a few working days. This minimizes the influence of atmosphere and ageing effects. In the following, standardized compression and bending tests as well as operating tests with cup geometries will be conducted under variation of tool material and slicing parameters. The obtained data from material characterization are then applied to a suitability assessment for sheet metal forming tools.

Flexural and compressive strength and modulus, as well as density, hardness, and surface roughness, are common material performance test values that are all considered relevant for the specific application of direct polymer additive tooling.

### 2.2. Material Characterization

The material properties identified as relevant for additively manufactured tooling from polymers are evaluated for four relevant materials.

Polylactide (PLA), as a standard, stiff 3D printing material, is chosen due to its former application for additive tooling [9,23];Polycarbonate (PC) is supposed to have strong material characteristics and was applied successfully in industrial sized additive tooling applications by [19];Polyamide (PA), a.k.a. Nylon, is a softer material but interesting for application in additive tooling due to its lower friction coefficients;PETG is included in the study as it is a good compromise of mechanical properties, availability, processability in AM, and recyclability.

ABS is a commonly used polymer in FFF. Due to its weak mechanical properties [23], it is excluded from this study and is therefore considered unsuitable for application in additive tooling. High performance polymers, such as polyether ether ketone (PEEK), are disregarded, as their material cost might sharply narrow the economic niche of additive tooling compared to conventional subtractive steel tool manufacturing practices and should be addressed in separate studies.

With regard to the tool manufacturing application, compressive and flexural properties are tested instead of standard tensile tests. In particular, these data are scarcely available in literature compared to tensile properties. Compressive and flexural material specimens are printed according to the technical standards EN ISO 604 and ISO 178 (also see Appendix A), respectively. For compression, cylindrical geometries with a diameter of 30 mm and a height of 20 mm are selected, whereas bars with a length of 100 mm and a cross section of 4 mm by 8 mm are manufactured for bending tests. As mentioned in the introduction, the mechanical parameters of a given polymer filament might significantly vary between manufacturers, and also between the used 3D printer. For this reason, the filament manufacturer and printer are kept constant in this study. Moreover, the processing parameters with regard to speeds and temperatures might influence the mechanical behavior of the material. However, they strongly relate to the overall construction of individual printers, e.g., hot end, heating, axis setup, etc. In the present study, they are kept to the manufacturer recommendations.

The slicing parameters, layer thickness, as well as perimeters are varied as they have to be defined by the user. They influence the mechanical stability and the surface quality of an additively manufactured object. Considering common FFF printing setups, the layer thickness is varied in between 0.1 and 0.3 mm, while the number of walls is set between 1 and 5. The corresponding specimens for are additively manufactured by FFF on Ultimaker “S5 Pro Bundle” machines using a 0.4 mm nozzle and the parameters summarized in Table 1. A solid (100%) infill was chosen, as the purpose of the study is to deliver mechanical material properties with maximized strength for tool applications. The printing orientation is perpendicular to the load because, from an application point of view, this orientation agrees to the manufacture of forming tools. The other printing parameters (i.e., nozzle temperature, bed temperature, and printing speed) are kept to the manufacturers’ suggestions in order to allow stable and comparable printing conditions. In the case of bed heating, the adjusted temperature of 80 degrees Celsius lies within the feasible ranges of all materials.

Before destructive mechanical testing, the specimens are evaluated for their surface roughness, using a Mahr Marsurf LD260, to obtain the surface roughness parameters *Rz* and *Ra*. In addition, hardness (Shore-D grade) was evaluated from the compression test cylinders by a digital indentation gauge with three repetitions each.

On a universal tensile testing machine type Zwick/Roell Z250, test assemblies for three-point bending and uniaxial compression are equipped. The compression and bending forces resulting from the experiments are continuously evaluated from the machine’s GTM load cell (series K, accuracy class 0.02%). For the compression tests, hysteresis loops are driven during the evaluation of compressive elastic modulus to eliminate settlements, gaps, and specimen surface unevenness. During elastic deformation, compression rates of 0.05 mm/s were achieved. Subsequently, the experiments run until compressive strain reaches 10%, cf. Figure 3. The obtained force over travel signals is compensated by a combined machine/assembly stiffness of 99.056 N/mm, which was evaluated prior by loading without specimen. For three-point bending, two supports are set into a lateral distance of 68 mm regarding their contact points towards the bar-shaped bending specimens. In between these supports, a wedge-shaped tool vertically applies the bending force. All material test raw data are provisioned in a data repository [39].

## 3. Material Test Results

The mechanical properties of specimens made by additive manufacturing with solid infill are evaluated with respect to compression and flexion with the intention to be applied as a tooling technique. Hence, material properties with the highest possible resilience are of interest. Looking at the results, the achieved strength values from all material and parameter variations are considered first, cf. Figure 4. The strongest combination with respect to both loading conditions is obtained by PLA with the finest layer resolution. Polycarbonate (PC) reaches similar strength but reveals no sensitivity against layer thickness. This observation could be interpreted as beneficial by means of production time, which generally decreases sharply with thicker layer heights. Likewise, PETG has shown a minor influence of layer height on strength, while nylon (PA) is strongly weakened by increasing layer height. At 0.3 mm layer height, PA delivers the weakest combination with respect to both bending and compression load situation.

For additive tooling, the elastic constants are of key interest to quantify local displacements of the tools under load and, thus, manufacturing precision. With regard to both loading conditions, bending and compression, the elastic moduli are evaluated from the tests during the initial loading phases and summarized in Figure 4. During compression tests, hysteresis loops are conducted, where compression is initially increased up to 75% of the material’s compression strength. Subsequently, force is decreased until 10% compression strength. From this point, the test is continuously performed up to 10% compression. From the slope of reloading the specimens within the loop, the elastic compression modulus *E*_c_ is evaluated where a linear regression behavior is assured by a coefficient of determination R^2^ > 0.9998 (also see Figure 3). The achieved elastic material constants under flexural and compressive load are displayed for all material and parameter combinations in Figure 5.

Corresponding to strength, the highest elastic constants are seen from the results of PLA, while PA delivers the weakest combination. Interestingly, the elastic moduli are influenced in a much lesser way by layer thickness, compared to the results on strength. Only PA shows a significant sensitivity with decreasing elastic moduli towards thicker layers. Similar to the strength tests, the results from PC show the second highest values.

Apart from these major results, the surface roughness (with regard to *Rz*) sharply increases towards thicker layers for PC (26 µm up to 125 µm for 0.1 mm to 0.3 mm layers) and PA (13 µm up to 63 µm), while PLA and PETG stay on comparatively smooth averages (22 µm and 28 µm, respectively). Surface hardness on the Shore-D scale could only be related with moderate correlation towards the results presented above. A weak influence of layer height and number of walls is obtained, except for PA, at the highest layers. Two groups of hardness are obtained with respect to the materials, with PLA and PC showing 81 degrees and 82 degrees of Shore-D hardness, respectively, on average, and PA and PETG revealing 76 degrees and 78 degrees, respectively, on average. For PA at 0.3 mm layers, a sharp drop of hardness towards 70 degrees is obtained.

Lastly, density *ρ* was evaluated from the 3D-printed compression test specimens because they show a higher volume to surface ratio than the bending bars. For all parameters adjusted, an average density of 1179 kg/m^3^ is obtained from the PC specimens. PETG weighs on average 1235 kg/m^3^, making it the densest material tested. The lowest density is obtained from the PA specimens having 1002 kg/m^3^ on average for 0.3 mm layer thickness and 1105 kg/m^3^ for the other parameter adjustments. Evidently, this sudden drop of density over layer height explains the weak material properties observed and might be related to a large number of voids resulting from this processing setup. As 0.4 mm nozzles were used throughout this study, the sudden drop in density and mechanical material properties over layer height might relate to the nozzle diameter. For PLA, a finely defined density variation is seen when varying layer thickness ranging from 1166 kg/m^3^ up to 1228 kg/m^3^ over decreasing layer height from 0.3 mm to 0.1 mm. PETG shows a disadvantageous balance of density ratio to its elastic constants and strength compared to the other materials. All numerical results obtained from the material tests are summarized in Table 2. The experimental history of all material tests is documented in a data repository [39].

In the subsequent cup-type drawing test, the strongest and weakest materials and parameter combinations are selected primarily according to compressive strength and modulus. Apart from the herein regarded deep drawing process, DPAT was successfully applied to a number of other forming processes, such as bending. Depending on the application case, the tested mechanical properties under bending load need to be taken into account to layout such tooling. Moreover, surface roughness is presented in Table 2 to assure a smooth surface of additively manufactured tools avoiding surface marks on the product. In the case of PC at a layer thickness of 0.3 mm, surface defects might occur depending on the sheet metal thickness of the product. Hardness and density are tested primarily for argumentative purpose, as weak mechanical properties might relate to a significantly low density, as in the case of PA at 0.3 mm. It is observed that the tested hardness shows mediocre correlation to the materials’ mechanical performance.

## 4. Use Case—Deep Drawing

Deep drawing is used as a forming process to apply the finding of the previous chapter to a real application tooling scenario. The wide application and the generally high tool loads make deep drawing a suitable use case for AT. Two different materials and their underlying parameter sets are used, showing the maximum span in compression strength of all tested material configurations: PLA (0.1 mm layer height and 1 wall) and PA (0.3 mm layer height and 5 walls). With respect to the properties’ flexural strength, surface roughness, compression modulus, and flexural modulus, the selected materials show the best and least performing configuration of all tested material configurations if minor variations that are caused by the change of wall numbers are neglected (cf. Figure 4 and Figure 5). The cup test, which is a commonly used deep drawing experiment, is implemented for the use case scenario. Similar to Bergweiler et al., the deep drawing specifications are set in such a way that the tools experience significant loads that are typical for deep drawing [17]. This can be achieved, for example, by setting the drawing ratio to the limit and by using a small die corner and punch nose radius. Table 3 summarizes all relevant deep drawing specifications for this use case.

The sheet metal material DC04, which is used for the experiments, is a cold rolled steel according to DIN EN 10130. The mechanical properties are listed in Table 4.

The experiments are executed using a three-part deep drawing toolset, consisting of a stamp, a die, and a blank holder. All three tools are printed for each material configuration with the same specifications. Shrinkage corrections are incorporated into the CAD geometry to enhance the initial accuracy of the printed parts. This iteration assures that identical geometrical prerequisites prevail in both cases of tooling material so that the influence of the tooling material configuration is analyzed exclusively from the results. In detail, the die diameter is increased by 0.2 mm for PLA, and by 0.5 mm for PA, respectively. Furthermore, the stamp is scaled-up by 0.5% in the direction of *x* and *y* and by 0.2% in *z*-direction, identically for both materials. Since the bank holder is a thin part with little material agglomeration, no shrinkage corrections are set for both materials. The knowledge of these shrinkage corrections comes from previous work and is not further detailed in the present study. Surface measurements of the initial tool geometries confirm the preset shrinkage corrections since the accuracy of the parts lays close to the respective layer height. Because PA material is printed with a layer height of 0.3 mm, whereas PLA is printed with a value of 0.1 mm, PA tools show slightly less accuracy, resulting in a maximum deviation span of −0.21 mm on the top surface of the stamp between those two materials.

The deep drawing experiments are carried out using a four-pillar die set, which is mounted on a single stroke press, having the specifications as described in [17]. A test series of 30 cups are drawn from each material configuration and subsequently measured using a GOM optical measurement device as described in [23]. The stamp and die surfaces of each material configuration are measured as well, using enlarging interval steps. The measuring points that are marked on the cup and tool of each graph of Figure 6, Figure 7, Figure 8 and Figure 9 are averaged around the circumference.

In order to compare the performance of both material configurations for the application scenario, two major key indicators come into play. The first one is the accuracy of the first drawn cup compared to the desired CAD geometry, which can be extracted from a surface comparison of the scanned part. This comparison gives an impression of the tolerances that can be potentially achieved by the underlying tooling configuration. The second indicator stands for the durability of the tools that is also linked to the accuracy of the drawn part or the wear on the tools over the course of a series. In the following context the term wear is used for the deviation change of the tool surface, which predominantly consists of plastic deformation. An analysis of different types of wear e.g., adhesive, abrasive, or erosive wear, are not further investigated and thus not part of the study. In forming tools, wear is caused by permanent frictional stress between the forming tool and the workpiece. According to the SOCIETY OF TRIBOLOGY, “wear is the progressive loss of material from the surface of a solid body caused by mechanical causes, i.e., contact and relative movement of a solid, liquid or gaseous mating body.” [40] (p. 108).

Figure 6 illustrates the accuracy of the first drawn cup and the repeatability of the drawn cups over the course of 30 drawing operations manufactured from the PA toolset. The left color map of Figure 6 shows the surface comparison of the first drawn cup to the desired CAD geometry. The global maximum and minimum deviation values are 0.88 mm and −1.48 mm, respectively. The color map in the middle of Figure 6 illustrates the surface comparison of the 30th drawn cup to the first drawn cup. Over the course of 30 drawing operations, the deviation values gradually increase, which can be seen in the graph of Figure 6, and reach a maximum span from −0.79 mm to 0.60 mm.

Figure 7 is structured in the same way as Figure 6 and shows the deviation values of the cups drawn from the PLA toolset. The first cup drawn from the PLA toolset is more accurate than the cup drawn from the PA toolset, reaching a global maximum and minimum deviation value of 0.36 mm to −0.87 mm, respectively. In addition, the test series shows a better reproducibility over the course of 30 drawing operations. A marginal decrease in precision can be noticed at a drawing step of around 18, leading to a maximum deviation span of −0.23 mm to 0.25 mm. The majority of the increased deviation span can be traced back to alignment errors during the surface analysis.

The color map of Figure 8 illustrates the surface comparison of the PA stamp after the 30th drawing operation compared to the initial stamp geometry right after printing. The deviation values can be interpreted as wear on the tool. The maximum and minimum deviation values can be seen around the punch nose radius. They are set between 0.20 mm and −0.45 mm, respectively, for the PA material. The change of tool wear is shown in the graphs for different measuring points for both material configurations. Deviation values increase for PA gradually until the 20th drawing operation and increase stronger until the end of the series. In contrast, the wear on the PLA stamp is much lower over the course of the series and lays in the range of alignment errors, thus it can be neglected.

The graphs and the color map of Figure 9 are structured in the same way as shown in Figure 8. However, the wear on the die is illustrated instead. The deviation values of the PA die are a bit less than on the PA stamp, ranging from −0.33 mm to 0.19 mm over the course of 30 drawing operations. In contrast, the wear on the corner radius of the PLA die is higher compared to the wear on the stamp, but still less than the wear on the PA die.

## 5. Discussion

Four polymers commonly used in FFF, namely PC, PLA, PA, and PETG, are considered for their use in direct polymer additive tooling (DPAT) in the present study. During additive fabrication of specimens intended for mechanical testing, the machining parameters are used according to the material manufacturers recommendations. Large differences between the mechanical parameters of these materials are observed with particular respect to compressive stiffness and strength. Initially, however, this study points out in a literature review that mechanical constants strongly vary across the different manufacturers. Therefore, the herein provided parameters are therefore intended to highlight the relative differences between the polymers rather than the absolute values, which might differ even more between the providers of a certain material than among different polymers of an identical provider.

Next to different polymers, the user-related FFF parameters, namely layer height and number of walls (or perimeters), are focused on the experimental design of this paper while the fabrication parameters relating to temperatures and speeds are kept constant. PLA and PA reveal a strong sensitivity to compression and bending related mechanical properties, while the properties of PETG are less affected. Considering the initially mentioned strong dependency of mechanical parameters from the manufacturer, a general recommendation towards thinner layers results. Compared with what is known from the literature on tensile, compression, or bending tests, the presented trend of increasing mechanical strength by lowering the layer thickness agrees with [30,32,34], for PLA. Interestingly, the mechanical integrity of PC is not degraded by raising the layer thickness, which might open up an economical gap as layer thickness directly influences the manufacturing time. Despite the large difference in mechanical parameters observed in the material test section, a low correlation of surface hardness is seen from the tests. Due to the discontinuous nature of additive manufacture, the testing needle of the Shore-D hardness measuring gauge might either penetrate on a tool path of the FFF process or in between two paths, which might be the reason for said lack of correlation. Clearly, this issue should be addressed and investigated in future research.

The mechanical parameters, with respect to bending and compression load tested in the first part of this study, are intended to relate to the quality which is expected from sheet metal drawing in DPAT. Consequently, the best and the worst material and layer thickness combination is applied to a use case in the second part of the paper to illustrate the span of manufacturing quality in a deep drawing operation. In addition to the corresponding fabrication parameters, which were also used for the mechanical testing specimens, a shrinkage compensation is conducted for the forming tools from PLA and PA. Resulting from the cup drawing experiments, a large difference between both tool materials is seen from the surface scans. The PA tools show higher product shape deviations and a considerable tool degradation after 30 strokes, while the shape deviation of the cups drawn on the PLA toolset lies within the margin of alignment errors during the drawing series. In addition, minor tool degradation was observed. Considering the mild drawing steel sheet material, DPAT shows suitability for sheet metal drawing of industrial products when using the herein recommended optimal parameters for strong polymer materials such as PLA. The findings of the parameter study can help the tool designer to manufacture stronger tools, which is mainly beneficial in two ways. First, an improved tool stiffness leads to sheet metal parts that are more accurate due to less elastic tool deformation. In addition, stronger tools are less prone to wear, which helps to manufacture more quantities that are acceptable in terms of part accuracy. Both advantages can lead to a wider range of application in industry for this tooling technique. Especially for prototypes and parts in small quantities, DPAT can be an economic way to manufacture sheet metal parts.

Forming of more complex geometries might result in higher local surface pressures on the tool surfaces and lead to higher permanent plastic surface deviations. In addition, high strength steels and higher sheet thickness could exceed the manufacturing capabilities of DPAT when using conventional polymers and might require stronger materials, e.g., polyether ether ketone (PEEK), fiber-reinforced polymers, or polymers with filler materials.

## 6. Conclusions

Utilizing direct polymer additive tooling (DPAT) in molding and sheet metal forming processes has been studied recently for the economical fabrication of small batch series. Up until now, exemplary materials have been tried out to demonstrate the feasibility of DPAT in particular for sheet metal forming, or tool running-in and degradation effects were investigated over production of various small series. However, a comprehensive comparison of materials and key parameters for DPAT is missing, as in comparison to the application range of additive manufacturing in general, DPAT materials need to be as strong as possible and are mostly made with solid infill.

In the present study, the mechanical properties from PC, PLA, PA, and PETG are evaluated under compression and bending, as these load cases are postulated as most relevant for tool manufacture. Moreover, density, hardness, and surface roughness are tested. Collectively, these parameters are considered relevant for the specific application of direct polymer additive tooling. While there is evidently a large difference in between the tested polymers results, literature indicates an almost as high dependency of material properties in between varying manufacturers for a given material. Therefore, the presented comparison aims for showing tendencies between polymers while keeping to a single material provider. An important aspect of the presented results is the dependency of the user-related manufacturing parameters in fused filament fabrication (FFF), namely layer thickness and number of walls. For layer thickness, the considered materials have shown a strong dependency towards increasing strength with thinner layers except for PC which revealed constant strength parameters. The number of walls was identified to be of subordinary relevance for material strength.

From the material tests, the best and worst case, PLA at 0.1 mm layer thickness and PA (Nylon) at 0.3 mm thickness, respectively, where applied to DPAT of a deep drawing tool set. A small series of 30 cups was formed on each tooling and subsequently evaluated by digital image correlation (DIC). A high influence of the tooling material results corresponds to the material properties evaluated in the preceding material test series. While the shape deviations of the cups formed on the PLA tooling lie between 0.36 mm and −0.87 mm, the tool degradation over the small batch series of 30 pieces lies within the margin of alignment errors of the DIC. For PA, the cups show deviations in between 0.88 mm and −1.48 mm. In addition, a considerable degradation of the PA tooling was observed from the small batch of 30 pieces, leading to an additional deviation of 0.8 mm. The shape deviations of the cup series are primarily linked to the considerably higher elastic modulus of PLA than PA as the stiffer tool material experiences less elastic displacement under the same load. The observed tool degradation relates to the differences in strength of both materials because higher strength leads to fewer local plastic deformations of the tooling, e.g., the fillet radius.

By the presented results, design, and layout, recommendations for DPAT are provided for sheet metal drawing operations. In particular, for small batch series, DPAT opens up economical chances compared to conventional tooling from steel. The results imply a tradeoff situation between a fast and economical tool manufacture by means of thicker layers and the strongest and most accurate combination given by thin layers, which results in longer production times.

## Figures and Tables

**Figure 1 polymers-14-01694-f001:**
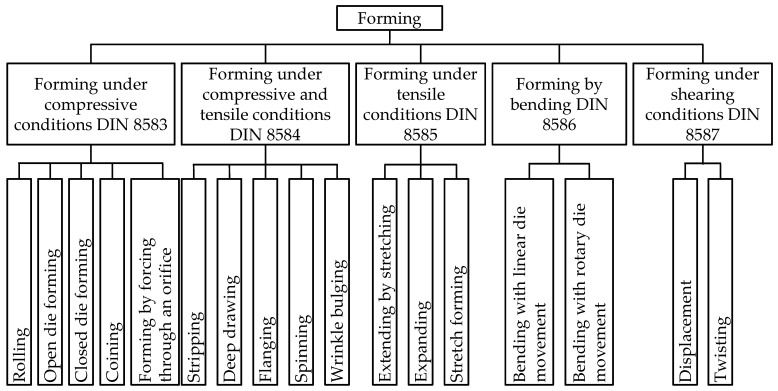
Classification of forming production processes according to DIN 8582, DIN 8585-1, adapted from Schuler GmbH [36] (p. 7).

**Figure 2 polymers-14-01694-f002:**
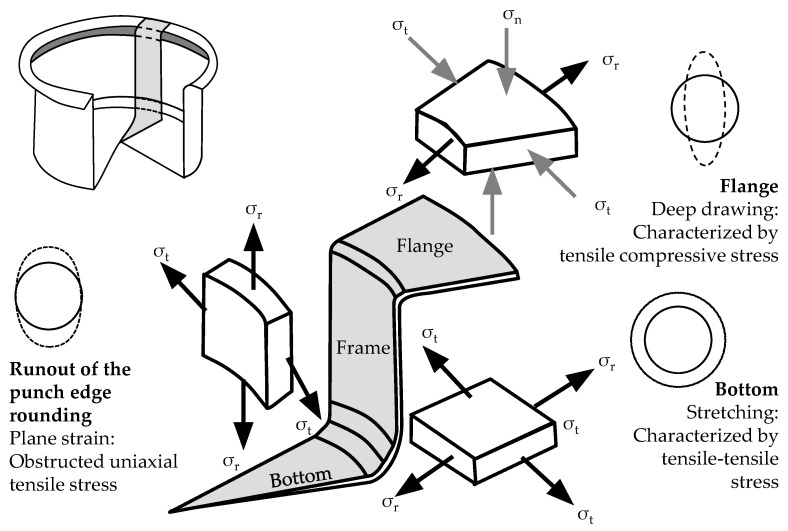
Stress ratios during deep drawing of a cup according to Doege (adapted from Kästle [37] (p. 16), Doege et Behrens [14] (p. 262)).

**Figure 3 polymers-14-01694-f003:**
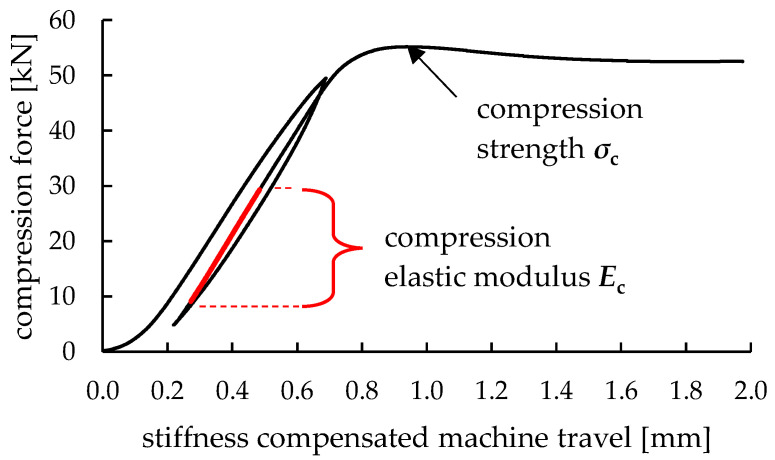
Machine stiffness compensated force-displacement curve of a compression test of a 3D printed solid cylinder from polycarbonate with a layer height of 0.2 mm and a single wall perimeter. A hysteresis loop is driven to eliminate any secondary influences, e.g., clearances. Indication of relevant areas for evaluation of mechanical material properties.

**Figure 4 polymers-14-01694-f004:**
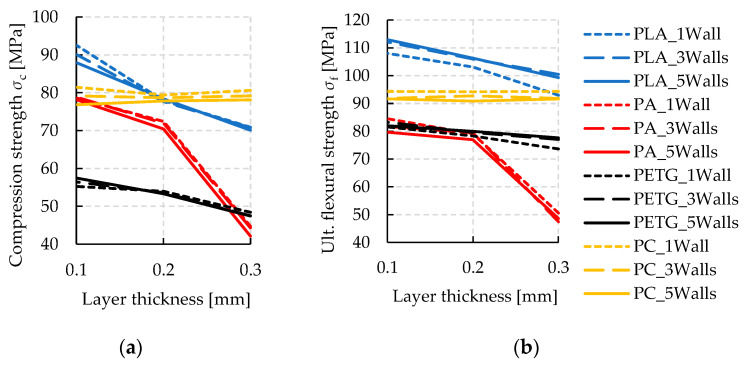
Resulting mechanical properties from the variation of materials and 3D-printing parameters. (**a**) Compression strength *σ*_c_ and (**b**) ultimate flexural strength *σ*_f_ results. A large influence of material and layer thickness is obtained, while the number of walls plays a subordinate role.

**Figure 5 polymers-14-01694-f005:**
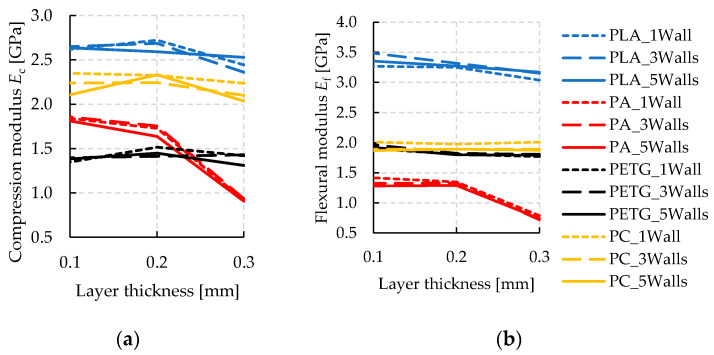
Elastic moduli resulting from the variation of materials and 3D-printing parameters. (**a**) Compression modulus *E*_c_ and (**b**) flexural modulus *E*_f_. A large material influence on elastic constants is obtained while the number of walls is of minor relevance. Only the elastic coefficients of PA are influenced significantly by layer thickness.

**Figure 6 polymers-14-01694-f006:**
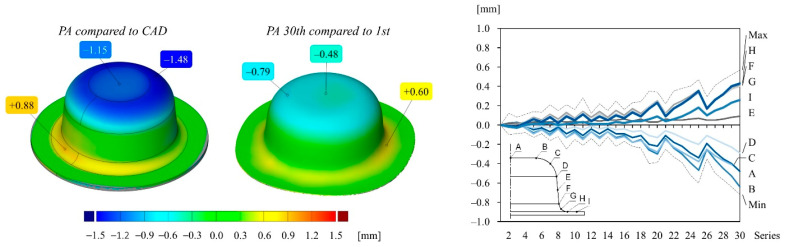
Color map and deviation values of different points of cups drawn from the PA tool: (**left**) first drawn cup compared to the CAD geometry; (**middle**) 30th drawn cup compared to the first drawn cup; (**right**) deviation values of different measuring points of each cup compared to the first cup over the course of 30 drawing operations. The drawing inside the diagram indicates the location of the measuring points on the cup surface.

**Figure 7 polymers-14-01694-f007:**
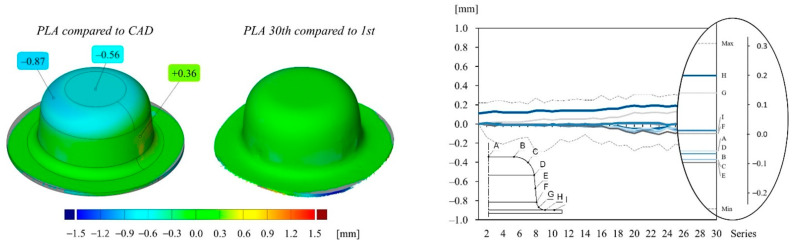
Color map and deviation values of different points of cups drawn from the PLA tool: (**left**) first drawn cup compared to the CAD geometry; (**middle**) 30th drawn cup compared to the first drawn cup; (**right**) deviation values of different measuring points of each cup compared to the first cup over the course of 30 drawing operations. The drawing inside the diagram indicates the location of the measuring points on the cup surface.

**Figure 8 polymers-14-01694-f008:**
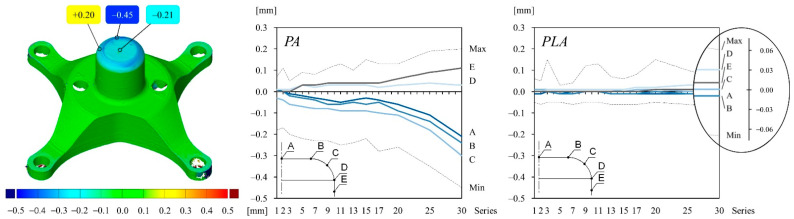
Deviation values (wear) of different measuring points on the stamp surface compared to the initial stamp geometry of both material configurations: (**left**) selected measuring points; (**middle**) deviation values of PA stamp over the course of 30 drawing operations; (**right**) deviation values of PLA stamp over the course of 30 drawing operations. The drawing inside the diagram indicates the location of the measuring points on the punch surface.

**Figure 9 polymers-14-01694-f009:**
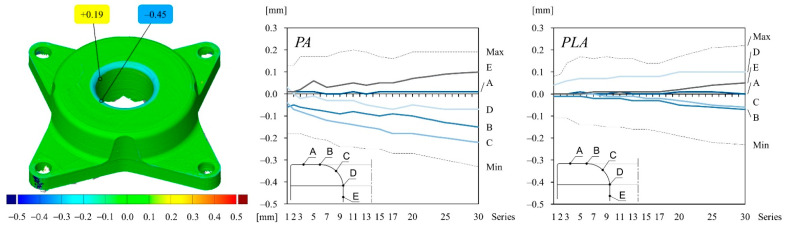
Deviation values (wear) of different measuring points on the die surface compared to the initial die geometry of both material configurations: (**left**) selected measuring points; (**middle**) deviation values of PA die over the course of 30 drawing operations; (**right**) deviation values of PLA die over the course of 30 drawing operations. The drawing inside the diagram indicates the location of the measuring points on the die surface.

**Table 1 polymers-14-01694-t001:** 3D printing parameters, applied to both specimen geometries (bending bar, cylindrical compression test spec.). Five repetitions were printed for each setup. While variations of layer thickness and number of walls are the main objective of this paper, nozzle and bed temperatures as well as printing speed are kept to the material manufacturers’ recommendations.

Material	Layer Thickness	Number of Walls	Nozzle Temperature	Bed Temperature	Printing Speed
	[mm]	[-]	[°C]	[°C]	[mm/s]
PLA	0.1, 0.2, 0.3	1, 3, 5	205	80	45
PC			270		
PETG			235		
PA			245		

**Table 2 polymers-14-01694-t002:** Average mechanical material parameters over FFF layer thickness. Main results of testing elastic bending modulus *E*_f_, ultimate bending strength *σ*_f_, elastic compression modulus after hysteresis *E*_c_, and compression strength *σ*_c_. Moreover, surface roughness *Rz*, surface hardness on Shore-D scale, and densities *ρ* were evaluated from the cylindrical specimens prior to compression tests.

Material	PC			PLA			PA			PETG		
Layer [mm]	0.1	0.2	0.3	0.1	0.2	0.3	0.1	0.2	0.3	0.1	0.2	0.3
*E*_f_ [GPa]	1.92	1.92	1.93	3.37	3.28	3.12	1.34	1.32	0.75	1.93	1.81	1.79
*σ*_f_ [MPa]	92	93	92	111	105	98	82	79	49	82	79	76
*E*_c_ [GPa]	2.23	2.30	2.12	2.63	2.67	2.44	1.84	1.71	0.92	1.38	1.46	1.39
*σ*_c_ [MPa]	79	78	79	90	78	71	79	72	44	56	54	48
*Rz* [µm]	26	60	125	14	26	26	13	23	63	15	26	30
Hardness (Shore-D)	82	82	82	84	80	81	79	77	71	79	78	78
*ρ* [kg/m^3^]	1186	1178	1172	1228	1202	1166	1114	1095	1002	1252	1239	1213

**Table 3 polymers-14-01694-t003:** Deep drawing specification for the use case.

No.	Feature	Symbol	Value (mm)
1	Blank diameter	*D*	53.1
2	Drawing ratio	*β*	2.125 (no unit)
3	Blank thickness	*S* _o_	1
4	Drawing depth	*h* _o_	15.77
5	Clearance	*C*	1.69
6	Die diameter	*D* _p_	28.38
7	Punch diameter	*D* _i_	25
8	Die corner radius	*R* _p_	2.5
9	Punch nose radius	*R* _i_	5

**Table 4 polymers-14-01694-t004:** Mechanical properties of DC04 as given in DIN EN 10130.

Material	Yield Strength	Ultimate Strength	Elongation at Break
DC04	210 MPa	270–350 MPa	38%

## Data Availability

The experimental history of all material tests is documented in a data repository: Frohn-Sörensen, P., Geueke, M., Löffler, B., Bickendorf, P. and Asimi, A. (2022), “Measurements of mechanical material properties under compressive and bending load of additive manufactured polymers”, Harvard Dataverse, 10 February, accessed on 31 March 2022, available at: https://doi.org/10.7910/DVN/1ARRM2.

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
