# Peer review of "Design for 3D Printed Tools: Mechanical Material Properties for Direct Polymer Additive Tooling"

_polymers, 2022, doi:10.3390/polym14091694_

Round 1

Reviewer 1 Report

The authors investigate the effects of parameters during direct polymer additive tooling (DPAT) process and discuss the mechanical properties of four polymer materials with different parameters. Analysis focused mostly on the deep-drawing cases with corresponding parameters and this work is appropriate for the audience of Polyers. The reviewer recommends the manuscript for publication.

1. Before the publication, the author should polish their manuscript carefully.

2. The authors should also discuss how their methods and study help the industry. More discussions are needed.

Author Response

Firstly, we would like to thank reviewer 1 for his/her time and consideration and the valuable feedback on our manuscript, which has been carefully processed and incorporated into this revision.

Reviewer 1

“The authors investigate the effects of parameters during direct polymer additive tooling (DPAT) process and discuss the mechanical properties of four polymer materials with different parameters. Analysis focused mostly on the deep-drawing cases with corresponding parameters and this work is appropriate for the audience of Polymers. The reviewer recommends the manuscript for publication.”

  • Before the publication, the author should polish their manuscript carefully.

Answer: We went the entire manuscript and thoroughly revised our language and formatting.

  • The authors should also discuss how their methods and study help the industry. More discussions are needed.

Answer: In the discussion, we added lines to the second to last paragraph to address this important issue.

Reviewer 2 Report

The idea of the article is original and can be useful for additive manufacturing. However, I have recommendations listed below.

The main changes authors should do are about the experimental characterisation of the materials.

Page 5, line 181: There is a problem with a reference;

Page 5, line 190: There is a problem with a reference;

Page 6, line229: Why Young’s modulus (elastic modulus) are not in GPa as usual?

Table 1: How were those printing parameters established? Are they optimized for all polymers considered?

Page 7, line 305: Authors should present some experimental compression test curves. What is the speed rate of the mechanical tests? What is the load cell capacity of the machine?

Page 10, Table2: Are Ef different of Eb on figure 4? Same question σf and Sc. Those parameters are not clearly defined.

Author Response

Firstly, we would like to thank reviewer 2 for his/her time and consideration and the valuable feedback on our manuscript, which has been carefully processed and incorporated into this revision.

Reviewer 2

„The idea of the article is original and can be useful for additive manufacturing. However, I have recommendations listed below.

The main changes authors should do are about the experimental characterisation of the materials.”

Answer: We corrected all mistakes in the material characterization paragraph with special care, also see answers below.

  • Page 5, line 181: There is a problem with a reference;

Answer: The missing reference led to Figure 2, the link was restored.

  • Page 5, line 190: There is a problem with a reference;

Answer: The missing reference led to Figure 2, the link was restored.

  • Page 6, line229: Why Young’s modulus (elastic modulus) are not in GPa as usual?

Answer: In this list, we changed the unit to GPa. Moreover, there was a problem in the Figure 5 (with reference to figure numbers of revised version) with the unit system, which has been adjusted.

  • Table 1: How were those printing parameters established? Are they optimized for all polymers considered?

Answer: An update to the describing text of the paragraph above as well as to the caption of this table are provided in the present manuscript revision.

  • Page 7, line 305: Authors should present some experimental compression test curves. What is the speed rate of the mechanical tests? What is the load cell capacity of the machine?

Answer: We added the characteristic force over displacement plot for an exemplary compression experiment and denoted the relevant areas, where mechanical parameters were examined (cf. new Figure 3). The speed rate is now provided and information on the utilized load cell is given in the revision.

  • Page 10, Table2: Are Ef different of Eb on figure 4? Same question σf and Sc. Those parameters are not clearly defined.

Answer: We updated our nomenclature and added the variables to every relevant caption for the sake of